# A Multi-Scale Spatial Difference Approach to Estimating Topography Correlated Atmospheric Delay in Radar Interferograms

**Zhigang Yu [1], Guoman Huang [2,\*], Zheng Zhao [2], Yingchun Huang [3], Chenxi Zhang [1] and Guanghui Zhang [1]**

[1]   College of Resources, Shandong University of Science and Technology, Taian 271000, China
[2]   Chinese Academy of Surveying and Mapping, Beijing 100830, China
[3]   Beijing Key Laboratory of Urban Spatial Information Engineering, Beijing 100038, China
\*    Correspondence: huang.guoman@casm.ac.cn

**Abstract:** The Interferometric Synthetic Aperture Radar (InSAR) has been widely used as a powerful technique for monitoring land surface deformations over the last three decades. InSAR observations can be plagued by atmospheric phase delays; some have a roughly linear relationship with the ground elevation, which can be approximated using a linear model. However, the estimation results of this linear relationship are sometimes affected by phase ramps such as orbital errors, tidal loading, etc. In this study, we present a new approach to estimate the transfer function of vertical stratification phase delays and the transfer function of phase ramps. Our method uses the idea of multi-scale spatial differences to decompose the atmospheric phase delay into the vertical stratification component, phase ramp component, and other features. This decomposition makes the correlation between the vertical stratification phase delays and topography more significant and stable. This can establish the correlation between the different scales and phase ramps. We demonstrate our approach using a synthetic test and two real interferograms. In the synthetic test, the transfer functions estimated by our method were closer to the design values than those estimated by the full interferogram–topography correlation approach and the band-pass filtering approach. In the first real interferogram, out of the 9 sub-regions corrected by the proposed method, 7 sub-regions were outperformed the full interferogram–topography correlation approach, and 8 sub-regions were superior to the band-pass filtering method. Our technique offers a greater correction effect and robustness for coseismic deformation signals in the second real interferogram.

**Keywords:** InSAR; interferogram; troposphere delays; multi-scale spatial difference; Sierra Nevada mountains; Menyuan earthquake

## 1. Introduction

The Interferometric Synthetic Aperture Radar (InSAR), with the advantages of satisfactory spatial resolution (decameters), comprehensive coverage (thousands of square kilometers), and competitive accuracy (millimeters to centimeters), has proven to be an effective means and method for ground deformations [1,2], volcanic deformation monitoring [3], seismic deformation inversion [4,5], surface building and infrastructure deformation monitoring [6–8], and landslide collapse disaster monitoring [9]. A Synthetic Aperture Radar (SAR) is a microwave sensor; its observations are frequently affected by atmospheric phase delays between the radar platform and the ground. According to previous studies, atmospheric phase delays significantly impact InSAR observations, which can lead to errors of 10 cm in deformations or hundreds of meters in elevations under particular circumstances [10].

Atmospheric phase delays include ionospheric delays and tropospheric delays. The spatial anisotropy of the ionosphere is very weak. Ionospheric delays can be omitted for short wavelength SAR data (C and X bands) [11]. Compared with the ionospheric delay,

the tropospheric delay is more significant, which is related to the atmospheric pressure, temperature, and water vapor content in the troposphere and cannot be neglected [12]. Spatially, atmospheric pressure and temperature change slowly, while water vapor content changes dramatically. Regarding vertical distribution, tropospheric delay can be divided into vertical stratification phase delays and turbulence delays. The vertical stratification phase delay is the static topographically correlated delay that results from different vertical refractivity profiles during the two SAR acquisitions. In mountainous areas, the vertical stratification delay has more impact on InSAR observations due to the large relief of the terrain. Turbulence delay results from turbulent processes in the atmosphere, i.e., turbulence in the atmospheric motion causes atmospheric delay errors in both the horizontal and vertical directions [13].

Since atmospheric phase delay is a critical error source in differential interferograms, there is a need to find the most straightforward and robust measures to quantify and mitigate the delay signal. One way to minimize the tropospheric phase delay is based on auxiliary data from sources such as the global navigation satellite system (GNSS) [14] or medium-resolution imaging spectrometry (MERIS/MODIS) [15,16] and other atmospheric reanalysis data (such as Modern Era Retrospective-Analysis for Research and Applications (MERRA), etc.) [17,18]. The monitoring network density of GNSS is sparse, and spatial interpolation processing is required between networks, which reduces the accuracy of atmospheric delay correction. MODIS data only works in the daytime and is easily affected by clouds. Atmospheric reanalysis data are difficult to synchronize with the time obtained from SAR images and have insufficient spatial resolution. The second method to mitigate the tropospheric delay is to average N-independent interferograms. This is due to the fact that the neutral atmospheric signals are uncorrelated over timescales longer than one day [19]. Hence, we can use the filtering techniques in the time series. The second method requires many interferogram pairs and considers that the atmospheric tropospheric delay is Gaussian, but this is not the case [20]. Another way is based on the spatial statistical characteristics of atmospheric phase delays to estimate and remove the effects of atmospheric delays. The efficacy of modeling-based approaches is still debated, especially the extent to which they consistently reduce or add noise to interferometric observations [21].

In mountainous areas or areas with sizeable topographic reliefs, the empirical function model between atmospheric phase delay and terrain elevation is established to mitigate atmospheric delay by analyzing the relationship between them. The atmospheric phase delay exists in multiple spatial scales. In addition, orbit errors cause a nearly linear ramp over the whole interferogram. Because of long-wavelength scale signals such as fault stable slip behavior [22], ocean tidal loading [23], and seasonal hydrological loading [24], the transfer function estimated by the empirical function model will have some deviation. In order to reduce the impact of such deviation, Lin et al. [21] and Shirzaei et al. [25] proposed techniques based on band-pass filtering and wavelets. These methods only consider the impacts of long-wavelength scale signals and ramps to improve the stability of the transfer function between vertical stratification delays and topographic elevation. Still, the transfer function parameters of long-wavelength scale signals and ramps are not estimated.

Our approach focuses on mitigating the effects of vertical stratification component delays in atmospheric phase delays and has a specific estimation ability for the impact of other linear long-wavelength scale signals and ramps. The approach proposed in this study is relatively simple and effective for correcting vertical stratification component delays without the need for other external auxiliary data. In our study, a phase component model is established based on the atmospheric phase delay characteristics of a single differential interferogram. For this purpose, a multi-scale spatial difference (MSSD) approach is proposed to estimate the transfer functions of vertical stratification component delays and ramp signals. Our method considers the spatial variability of both elevation and horizontal space of atmospheric phase delays. The MSSD method can estimate the vertical stratification component delays more significantly and stably. We tested our approach with 160 synthetic interferograms containing different ramps and turbulence. Using the synthetic

experimental results, we show how our method is insensitive to linear long-wavelength scale signals. Next, we demonstrate our approach with examples from the Sierra Nevada Mountains in the western United States and the Menyuan earthquake in Qinghai Province, China. In both examples, compared with the full interferogram–topography correlation approach and band-pass filtering approach, the interferogram corrected by the MSSD approach shows more improvement in sub-regions. The ramp of the phase is also significantly modified. We show that most of the remaining signals are mainly due to turbulence delays, which require more complex correction methods than those described here.

The rest of this article is organized as follows. Section 2 introduces the model for decomposing the atmospheric phase and the proposed approach for estimating the transfer function of vertical stratification phase delays and the transfer function of phase ramps. Sections 3 and 4 describe a synthetic test and two real interferograms to analyze the reliability of the proposed method, respectively. Finally, Section 5 discusses the proposed approach and concludes this article.

## 2. Model and Estimation Approach

### 2.1. Model

The atmospheric phase delay has a multi-scale spatial distribution, with some being the component with a larger wavelength scale, some being the turbulence component and some vertical stratification component delays. Our model decomposes the atmospheric phase delay into three major features: the vertical stratification component delay, the long-wavelength scale signal and ramp, and the turbulence and noise signal. The transfer function and bias term of the vertical stratification component delays in the atmospheric phase delay is stable, not affected by spatial changes, and approximate to a simple linear relationship [26,27]. The phase of long-wavelength scale signals changes approximately linearly in the horizontal space and does not change with topography. We also considered the influence of phase ramps across the scene. Here, we combine the linear long-wavelength scale and ramp signals into a "ramp". Turbulence signals are correlated in a short range (a few km) [28]. Furthermore, the noise signals have random characteristics. Hence, the atmospheric phase delay can be expressed as follows:

$$\phi = \begin{cases} \phi_{trop} = K_1 h + \phi_c \\ \phi_{line} = K_2 x \\ \phi_{other} = \phi_{tur} + \phi_{noise} \end{cases} \tag{1}$$

where $K_1$ and $\phi_c$ are the transfer function and bias term between the vertical stratification component delays $\phi_{trop}$ and the topographic elevation $h$ in the interferogram, respectively. $K_2$ is the transfer function of the ramp component delays $\phi_{line}$, $x$ is the position, and $\phi_{other}$ includes turbulence signals $\phi_{tur}$ and noise signals $\phi_{noise}$.

### 2.2. Estimation Approach

In the interferogram, the phase difference between the two points $i$ and $j$ in the ramp gradient direction is as follows:

$$\phi_{ij} = \begin{cases} \Delta\phi_{trop} = K_1 h_{ij} \\ \Delta\phi_{line} = K_2 S \\ \Delta\phi_{other} = \Delta\phi_{tur} + \Delta\phi_{noise} \end{cases} \tag{2}$$

where $\phi_{ij} = \phi_j - \phi_i$ is the phase difference between points $i$ and $j$; $h_{ij} = h_j - h_i$ is the topographic height difference between points $i$ and $j$, which can be calculated from the digital elevation model (DEM). $S = x_j - x_i$ is the distance between points $i$ and $j$ in the ramp gradient direction, which is called the scale factor of difference. $\Delta\phi_{other}$ is the phase difference of the $\phi_{other}$. After the difference, $\Delta\phi_{trop}$ is linearly correlated with the topographic height difference of the two points $i$ and $j$. $\Delta\phi_{line}$ is linearly correlated with the scale factor $S$. When $S$ is fixed, $\Delta\phi_{line}$ is a constant. In addition, $\phi_{tur}$ can weaken each other

in the case of short-distance differences since the atmospheric phase delays have a strong correlation in a short range [19]. The noise component has random characteristics, and its average value is approximately zero. Here, we omit $\Delta\phi_{other}$ and simplify Equation (2) as follows:

$$\phi_{ij} = K_1 h_{ij} + K_2 S \tag{3}$$

Equation (3) shows that a new linear equation can be obtained after the difference of interferogram, where $K_1$ is the transfer function and $K_2 S$ is the bias term. In this equation, the phase difference $\phi_{ij}$ and the topographic height difference $h_{ij}$ are linearly related, and the transfer function $K_1$ of this linear relationship is the same as that of the atmospheric phase delay and topographic height relationship of the original interferogram. In Equation (3), $K_1$ is independent of the spatial scale factor $S$, while the bias term $K_2 S$ is proportional to the spatial scale factor $S$. Therefore, $K_2$ remains unchanged before and after the difference. We use Equation (3) to fit the phase difference $\phi_{ij}$ and topographic height difference $h_{ij}$ on the difference of multiple spatial scale factors to obtain $K_1$ and $K_2 S$ and then fit $K_2 S$ of different spatial scale factors to estimate $K_2$. The transfer functions $K_2$ of the ramp in different directions are unequal, and the absolute value in the ramp gradient direction is the largest. Here, we estimate $K_2$ through eight directions (i.e., azimuth angles 0°, 45°, 90°, 135°, 180°, 225°, 270°, and 315°, respectively) with an interval of 45°. The maximum absolute value is taken as the transfer function of the ramp, and the corresponding direction is taken as the ramp gradient direction.

Due to the symmetry of the eight directions, it is only necessary to estimate $K_2$ in the four directions of 0°, 45°, 90°, and 135°. Under these circumstances, the difference between the ramp gradient direction and the estimated direction is 22.5° at most, and the resulting deviation is $K_2(1 - \cos(22.5°)) \approx 0.08 K_2$, which can be ignored (See the Appendix A for the calculation process). To decrease the effect of turbulence on the estimation, we adjusted the difference scale factor to no more than 5 km.

Figure 1 shows the results of the partial difference processed by our approach. Due to the influence of atmospheric turbulence and ramp, it is difficult to observe the relationship between atmospheric phase and topography from the original interferogram. The $K_1$ value calculated using the full interferogram–topography correlation is 1.81 rad/km, which considerably differs from the designed value of 2.5 rad/km (Figure 1). The correlation coefficient R of phase difference and topographic height difference is about 0.6, while the R of phase and topography in the original interferogram is 0.34. An evident correlation can also be observed between phase difference and topographic height difference. Compared with the scatter plot (phase vs. topography) of the original interferogram, the difference scatter plot (phase difference vs. topographic height difference) is more concentrated, and the estimated $K_1$ value is close to the design parameters.

Figure 2 depicts the correlation coefficient of phase difference (R), the topographic height difference of multiple spatial scales in four directions, and the estimated values of $K_1$ and $K_2 S$. With the increase in the difference scale, $K_2 S$ in the four directions keeps a linear change, among which the absolute gradient of 0° azimuth is the largest. This is consistent with the actual parameter (see Section 3 synthetic test). Even in other directions, the estimated $K_1$ value after the difference is closer to the actual parameter than the full interferogram–topography correlation approach. Figure 2 also shows that the correlation coefficient generally presents a gradually decreasing trend with the increase in the difference scale, and the estimated $K_1$ moderately deviates from the actual value. As a result, when the correlation coefficient is at its highest, $K_1$ may be used as the final estimation. However, due to the influence of unwrapping errors and other errors, in the case of real interferograms, the correlation coefficient R is not the maximum when the difference scale is small (see Section 4.1. Sierra Nevada Mountains). Finally, the transfer function estimated by the minimum scale (i.e., the distance between two adjacent pixels) is taken as the final $K_1$.

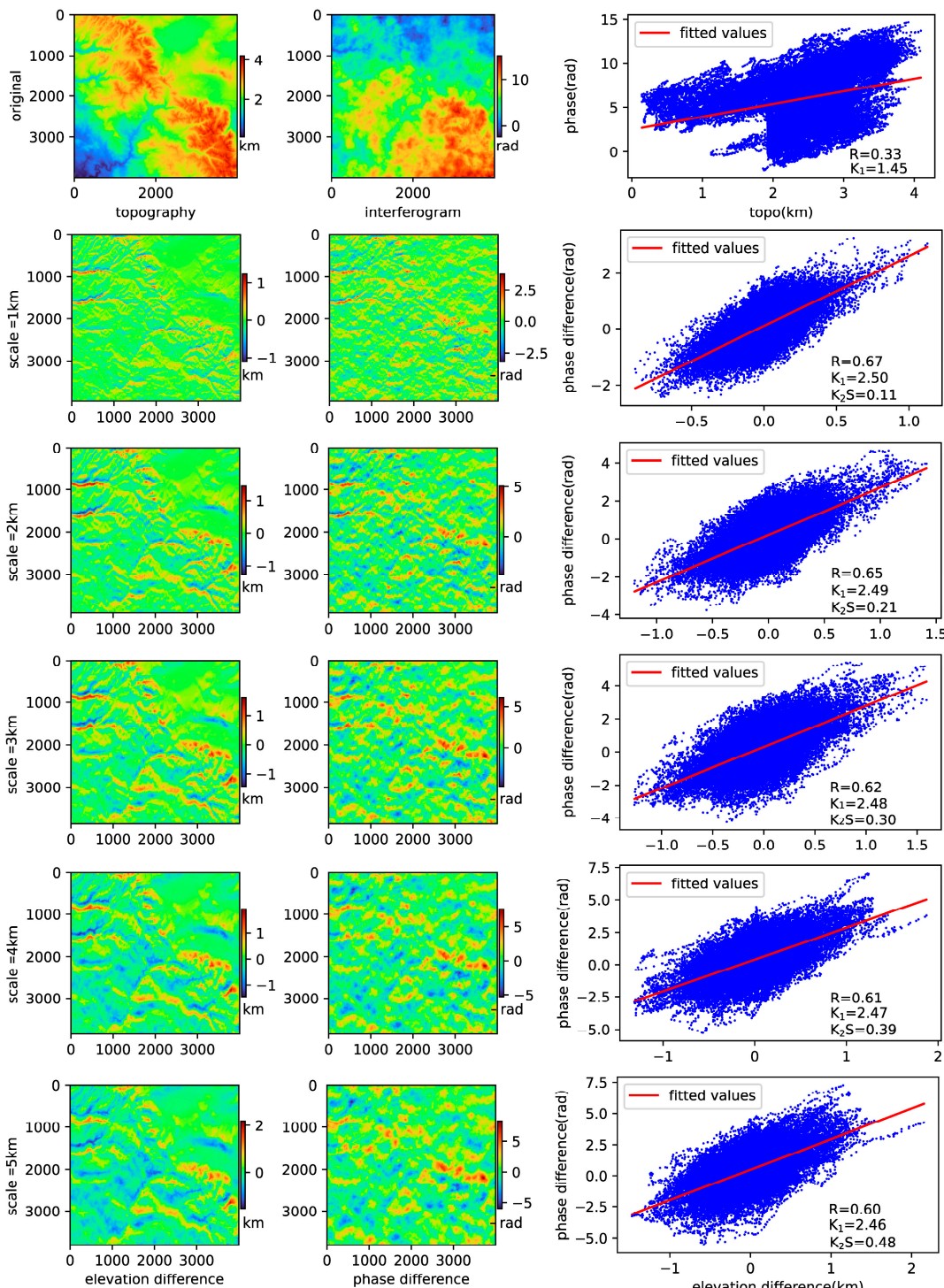

**Figure 1.** Original (**top**) and differential topography (**first column**) and interferogram (**second column**) with scale factors of 1 km, 2 km, 3 km, 4 km, and 5 km, respectively. The last column is the scatter plots of phase differences and topographic height differences. The scatter plots are diluted 300 times, and the direction of difference is azimuth 0°. The estimated values of $K_1$, $K_2S$, and correlation coefficient R for each scale factor are shown at the bottom right corners of the scatter plots. The final estimate values of $K_1$ and $K_2S$ are 2.50 rad/km and 0.1 rad/km, respectively, which are equal to the values set in the synthetic test (2.5 rad/km and 0.1 rad/km). In comparison, the $K_1$ value calculated using full interferogram–topography correlation is 1.81 rad/km.

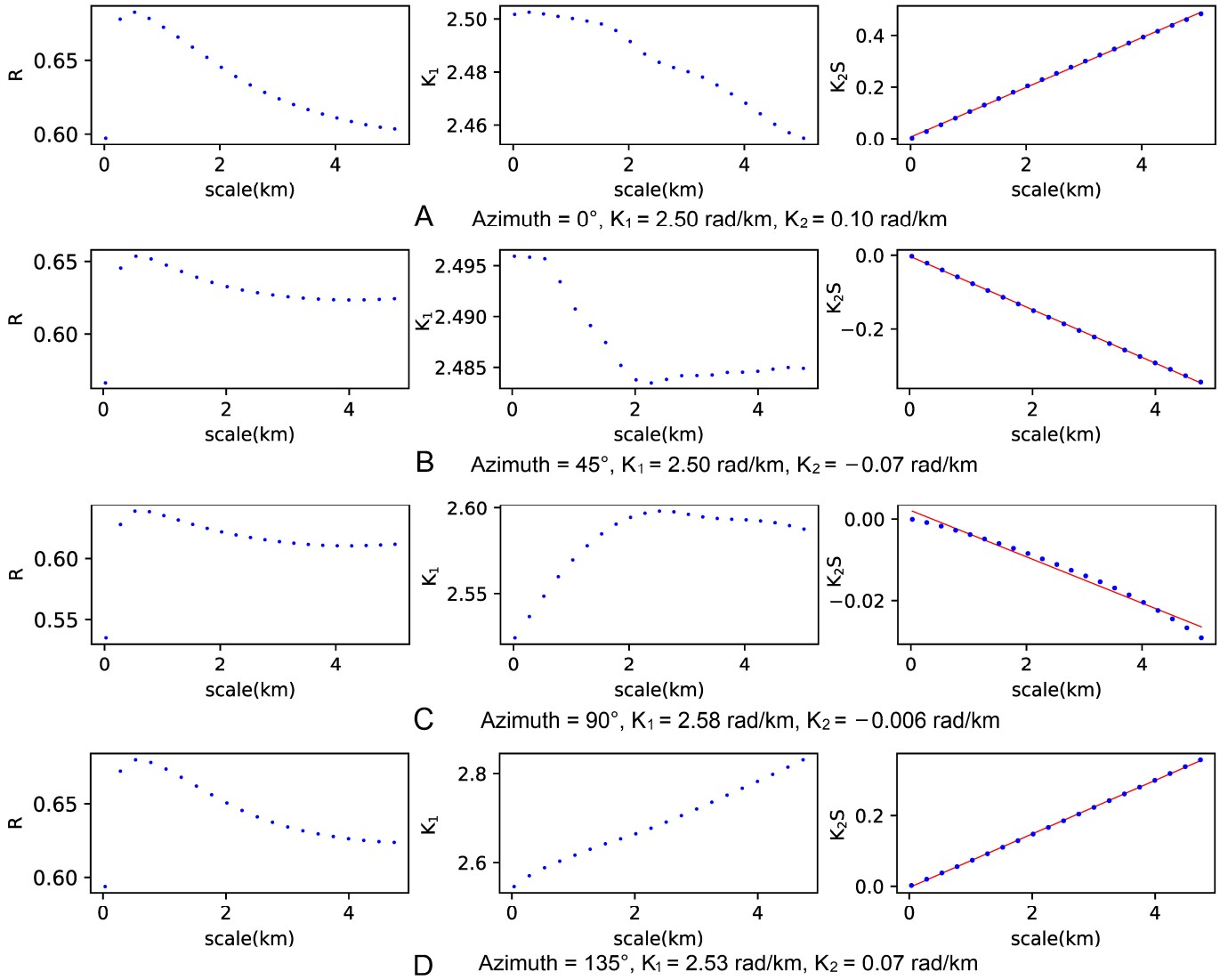

**Figure 2.** Values of the correlation coefficient R (column 1), $K_1$ (column 2), and $K_2S$ (column 3, blue dots, the red line is the fitting line of the blue dots) after the difference of multiple scales in four directions; the final $K_1$ is estimated according to the proposed method. The interval of the difference scale factor is 0.25 km, and the first scale factor is 0.025 km. The maximum absolute value of $K_2$ is 0.10 rad/km (**A**), the corresponding azimuth is 0°, and the related $K_1$ is 2.50 rad/km. The $K_2$ values of other azimuth (**B**–**D**) are all less than 0.1 rad/km.

## 3. Synthetic Test

In our experiment, we selected the DEM of the Sierra Nevada Mountains in the western United States to synthesize the interferogram. An ancient saline lake named Mono Lake is located at the eastern edge of the Sierra Nevada. The altitude of this area ranges from 140 m to 4.13 km. The topography is flat in the southwest, with a mountain range extending northwest to southeast in the middle (Figure 3). The components considered in the synthetic interferogram include vertical stratification component delays, turbulence, and ramp signals. We also simulated a simple deformation to verify the robustness of our method to deformation signals. In this section, we examine the efficacy of the suggested method by varying the turbulence signal intensity, amplitude, and ramp direction.

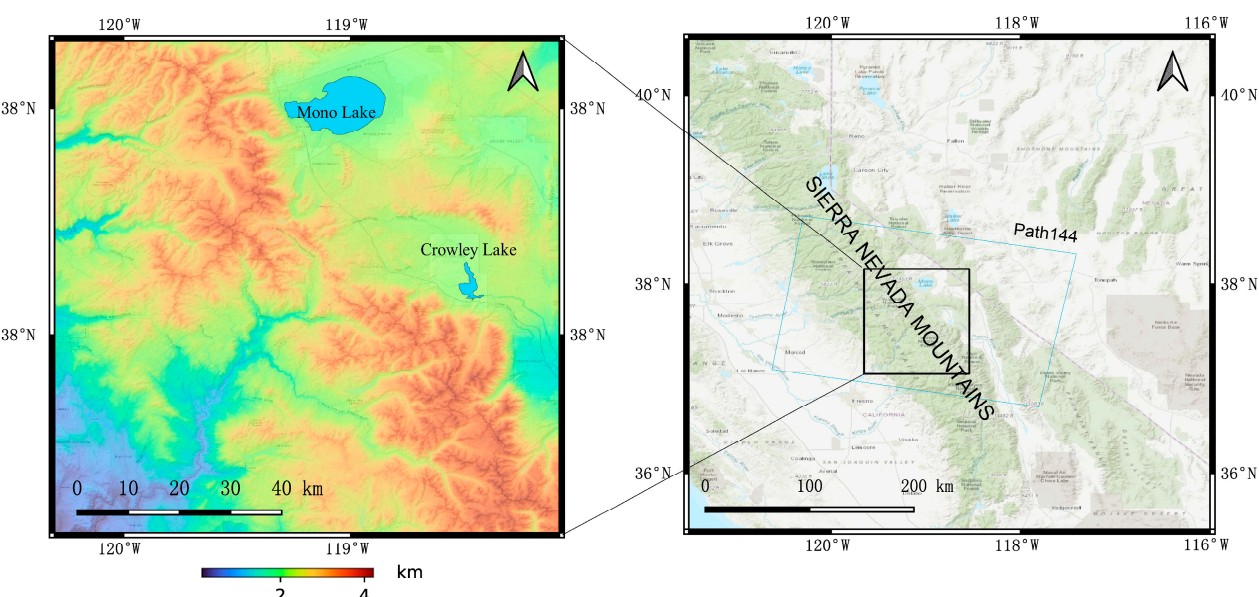

**Figure 3.** Reference map of the Sierra Nevada Mountains with maximum elevation up to 4.13 km. The blue frames are the coverage of the Sentinel-1A data.

As for turbulence signals, we chose the modified von Karman (MVKS) phase turbulence model. The power spectrum density of the von Karman Spectrum (also called the modified von Karman spectrum) is given by [29]:

$$\Phi_n(k) = 0.033 C_n^2 \frac{\exp(-\frac{k^2}{k_m^2})}{(k^2 + k_0^2)^{\frac{11}{6}}}, 0 \leq k \leq \infty \tag{4}$$

where $C_n^2$ is the medium structure parameter, $k_m = 5.92/l_0$ is an equivalent wavenumber associated with the inner scale $l_0$, $k_0 = 2\pi/L_0$ is a wavenumber related to the outer scale $L_0$, and $k$ is the unbounded non-turbulent wavenumber in the medium. In the above equation, $\Phi_n(k)$ represents the so-called power spectral density (PSD) of the refractive medium index. In standard atmospheric turbulence literature, it is known that the numerical ranges of the structure parameter $C_n^2$ define strong, intermediate, and weak turbulence. While the above represents the necessary values of the structure parameter defining a specific turbulence regime. It also needs to specify the so-called Fried parameter $r_0$ corresponding to the chosen $C_n^2$. This is obtained via the following formula [30]:

$$r_0 = 0.185 \left[ \frac{4\pi^2}{k^2 L C_n^2} \right]^{3/5} \tag{5}$$

where $L$ is the propagation distance. The Fried parameter means the diameter of a circular area, over which the root mean square (RMS) of the wavefront aberration due to passage through the atmosphere, equals 1 rad.

The second parameter is the amplitude of the ramp. Here, we assume a ramp that varies bilinearly in space. A small ramp parameter is set to 1 rad, equivalent to 0.01 rad/km, close to the amplitude of tectonic signals [21]. A large ramp is set to 10 rad, equal to 0.1 rad/km. The gradient direction of the ramp is set to 0° and 112.5°, respectively. The last parameter is the Fried parameter $r_0$ of turbulent signals, which is set to 5 km and 50 km, corresponding to turbulence amplitude of about 9 and 1.5 rad, respectively. The other parameters are kept constant during the simulation process. These parameters consist of the number of sample points (grid resolution) = 4000 × 4000, the size of grid = 25 m × 25 m, the inner scale $l_0$ = 10 m, and the outer scale $L_0$ = 30 km, respectively. The transfer function of vertical stratification component delays and topographic elevation is set to 2.5 rad/km. In the synthesis test, we

also construct the deformation signals by using a point source of inflation in an elastic half space [31] to demonstrate the robustness of our proposed approach. The surface deformation generated by the tectonic signal is 7.57 rad, and the range is about $2000 \times 2000$.

In total, we generated eight categories of interferograms regarding different Fried $r_0$ and ramp parameters, and each category corresponded to 20 interferograms. We expressed delay results in terms of phase (unit rad). Figure 4 illustrates one realization of our synthetic interferograms.

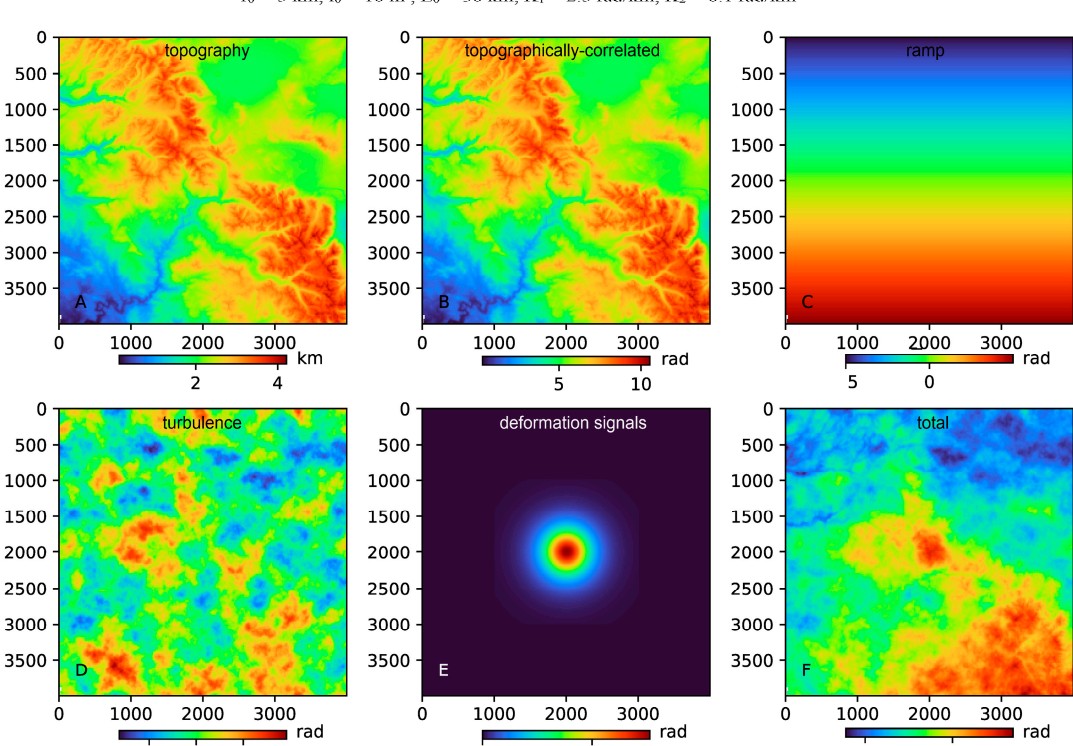

**Figure 4.** A schematic description of the construction of the synthetic interferometry: (**A**) Topography; (**B**) Topographically correlated tropospheric delays; (**C**) Large bilinear ramp are computed as described in the text; (**D**) Turbulent signals; (**E**) deformation signals that we project them to phase and combine them to form the (**F**) final synthetic interferogram. In this example, the Fried parameter $r_0$ is 5 km, the inner scale $l_0$ is 10 m, the outer scale $L_0$ is 30 km, the $K_1$ is 2.5 rad/km, and the ramp amplitude ($K_2$) is 0.1 rad/km.

Next, we estimated $K_1$ and $K_2$ values from each synthetic interferogram using the MSSD approach. For comparison, we also estimated the $K_1$ from the full interferogram–topography correlation approach and band-pass (BP) filtering approach [21] (Figure 5). We used the average value and measures of dispersion (standard deviation) to evaluate the results of the three processes (Table 1) regarding different parameters. Different $r_0$ could be used to compare the impact of turbulence on the transfer functions estimated by the three approaches. The dispersion of $K_1$ predicted using the full interferogram–topography correlation technique was more significant in solid turbulence ($r_0 = 5$km) than in weak turbulence (Figure 5A 0.200 vs. Figure 5E 0.031, Figure 5B 0.192 vs. Figure 5F 0.032, Figure 5C 0.268 vs. Figure 5G 0.033, and Figure 5D 0.252 vs. Figure 5H 0.024). Although the same situation exists for the $K_1$ value estimated by the MSSD method (Figure 5A 0.016 vs. Figure 5E 0.002, Figure 5B 0.013 vs. Figure 5F 0.002, Figure 5C 0.016 vs. Figure 5G 0.003, and Figure 5D 0.019 vs. Figure 5H 0.003), the dispersion of the $K_1$ value estimated by MSSD was significantly smaller than that of the $K_1$ value estimated by the full interferogram–topography correlation approach. The results of the BP filtering approach were closer

to those of the MSSD approach. It can also be concluded from Figure 5 and Table 1 that the $K_1$ value calculated by the full interferogram–topography correlation had an overall deviation. The size of the deviation was proportional to the amplitude of the ramp. In the study area of the test, the land surface showed a general trend in high in the northeast and low in the southwest. We believe that such trend in land surface slope and the ramp affected the $K_1$ value calculated by the full interferogram–topography correlation. Both MSSD approach and BP approach were less affected by the ramp, and the estimated $K_1$ value was more stable and closer to the design value of the experiment. The ramp direction in Figure 5 BDFH was 112.5°, having a 22.5° difference from the estimated direction, while the predicted $K_1$ values remained relatively accurate. The $K_2$ values have a relatively small deviation in Figure 5B,F (0.095 and 0.093, respectively), which was consistent with the design of our approximate calculation.

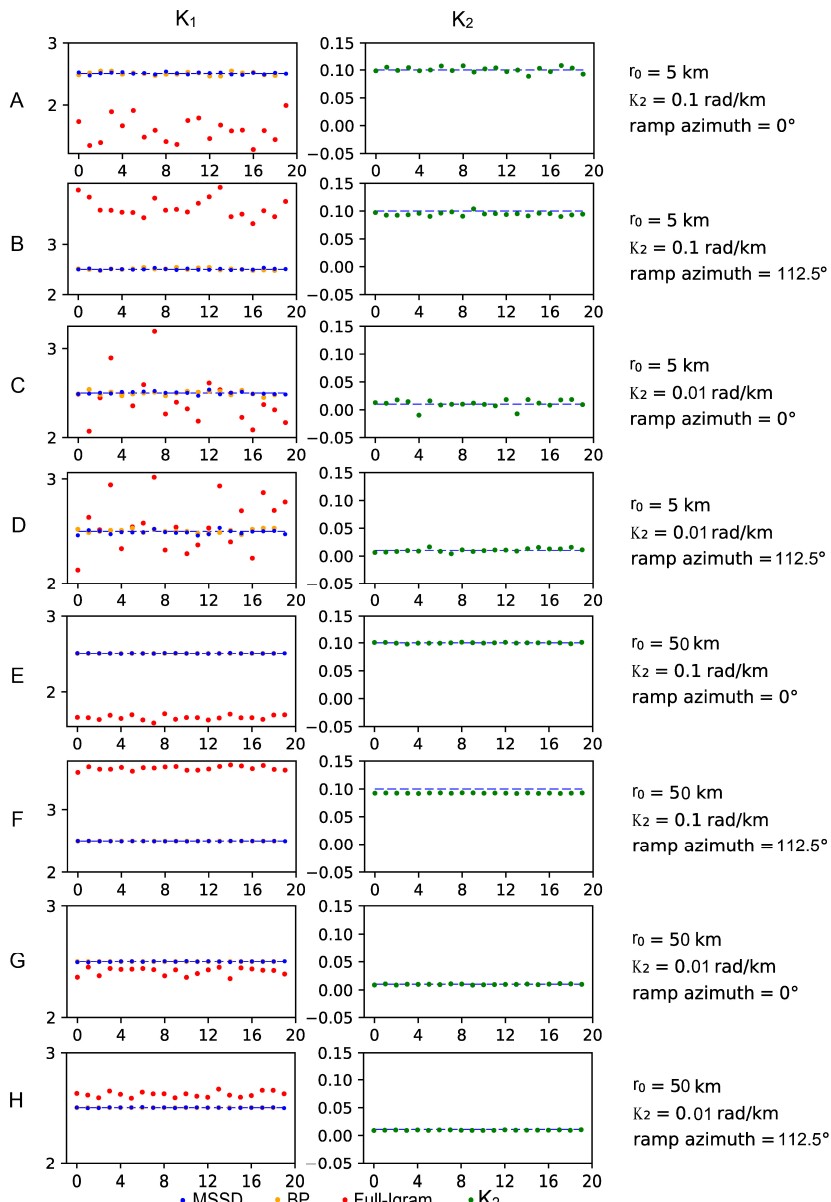

**Figure 5.** Comparison of the transfer functions estimated by using full interferogram–topography correlation, BP, MSSD, and the $K_2$ estimated by our method. There are 20 realizations of synthetic interferograms in each plot, with different turbulence signals, amplitudes, and ramp directions. The input $K_1$ is 2.5 rad/km (first column, blue dashed lines). The input $K_2$ is 0.1 rad/km (**A,B,E,F**) and 0.01 rad/km (**C,D,G,H**) respectively (second column, blue dashed lines).

**Table 1.** Statistical values of transfer functions estimated by three approaches: average value (AVG) and standard deviation (S.D.).

| Group | | A | B | C | D | E | F | G | H |
|---|---|---|---|---|---|---|---|---|---|
| $K_1$ (MSSD) | AVG | 2.503 | 2.505 | 2.500 | 2.492 | 2.500 | 2.499 | 2.500 | 2.500 |
| | S.D. | 0.016 | 0.013 | 0.016 | 0.019 | 0.002 | 0.002 | 0.003 | 0.003 |
| $K_1$ BP | AVG | 2.499 | 2.507 | 2.498 | 2.507 | 2.500 | 2.500 | 2.500 | 2.500 |
| | S.D. | 0.025 | 0.019 | 0.023 | 0.019 | 0.000 | 0.000 | 0.000 | 0.000 |
| $K_1$ (Full-Igram) | AVG | 1.603 | 3.734 | 2.426 | 2.569 | 1.665 | 3.656 | 2.413 | 2.620 |
| | S.D. | 0.200 | 0.192 | 0.268 | 0.252 | 0.031 | 0.032 | 0.033 | 0.024 |
| $K_2$ (MSSD) | AVG | 0.101 | 0.095 | 0.011 | 0.011 | 0.100 | 0.093 | 0.010 | 0.010 |
| | S.D. | 0.005 | 0.003 | 0.008 | 0.003 | 0.001 | 0.000 | 0.001 | 0.000 |

The results showed that our approach gave a stable estimate of $K_1$ and $K_2$ values regardless of the turbulence signal strength, the amplitude, and the direction of the ramp. To summarize the results from the synthetic test, our approach provided a more robust way to estimate the transfer function. This method was less sensitive to phase ramps and more adaptable to turbulence, resulting in a more accurate calculation of $K_1$ in the presence of orbital inaccuracy or significant turbulence signals. There may have been a slight fluctuation of transfer functions depending on the characteristic amplitude of turbulent signals.

## 4. Correcting Real Interferogram

We tested the MSSD approach in two study areas. Our first example was in the Sierra Nevada Mountains, California. It was assumed that there was no deformation signal in this area during the observation period, only the turbulence signal. We used this example to emphasize the effectiveness of our algorithm in the presence of large-amplitude turbulence noise. Our second example was the 2016 Menyuan earthquake (Qinghai Province, China). This example presented a relatively simple tectonic source combined with complicated atmospheric turbulent signals. Next, we tested the robustness of $K_1$ and $K_2$ by removing the tectonic signals from the interferogram.

### 4.1. Sierra Nevada Mountains

We apply the MSSD approach to a 12-day interferogram generated by SAR images acquired in descending track of the Sentinel-1A TOPS (Terrain Observation with Progressive Scans) mode (path 144) over the Sierra Nevada Mountains in the western United States (Figure 3). The master and slave images of this interferogram were acquired on the 9 June 2020 and 21 June 2020. Therefore, we assumed no deformation during this period and the phase change was mostly due to atmospheric delay. SRTM DEM with a 1-s resolution and AUX_POEORB precise orbits with an accuracy of 5 cm were used to flatten the interferogram. The interferogram was obtained by the InSAR Scientific Computing Environment (ISCE) [32]. The interferogram was multi-looked by factors of 19 and 7 along range and azimuth to improve the coherence and reduce the unwrapping error. The unwrapped interferogram was obtained by applying the SNAPHU approach [33]. After masking the low-coherence points (i.e., coherence less than 0.3), the interferogram was obtained, as shown in Figure 6B, which clearly indicates that there was a complex phase. Furthermore, there was a ramp from northwest to southeast, which was evidently inconsistent with the terrain of the region. Although we adopted large multi-look parameters to reduce the unwrapping errors and mask the interferogram with coherence of less than 0.3, there were still some noises and unwrapping errors in the interferogram. These errors were due to the influences of lake, vegetation, and layover in the study area, which may lead to incoherence or unwrapping errors [34], as shown in Figure 6C–E.

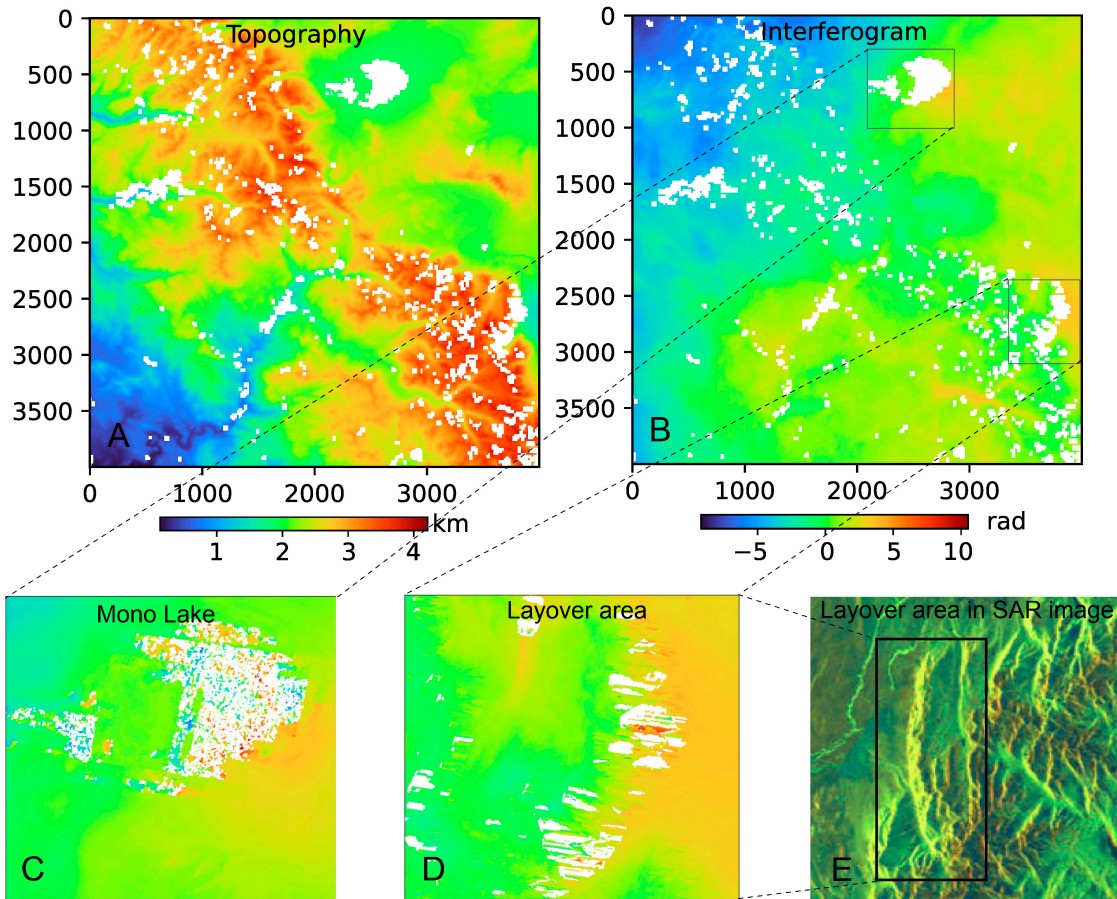

**Figure 6.** (**A**) The topography of the Sierra Nevada mountains; (**B**) the original interferogram; (**C**) the Interferogram of the Mono Lake area (Despite the mask processing, a large number of noise signals remain in this area); (**D**) the layover area in the interferogram; (**E**) the layover area in SAR image.

Figure 7 depicts the correlation coefficient R of phase difference and topographic height difference in four directions, as well as the estimated values of $K_1$ and $K_2S$. It shows that the estimated $K_1$ of the minimum difference scale (0.035 km, azimuth = 135°) is about $-0.2$ rad/km in four directions. With the increase in the difference scale, the estimated $K_1$ absolute value shows a gradually increasing trend, which is close to the situation of the synthetic test. The absolute value of correlation coefficient R gradually increases with the increase in the difference scale, which is different from the condition of the synthetic test.

Figure 8 shows the scatter plots under various difference scales at 135°. When the difference scale is 0.035 km (Figure 8A), the phase difference is concentrated between the fitted value $\pm1$ rad within the 95% confidence band, and the elevation difference is concentrated in the range $\pm0.05$ km. The correlation coefficient R is significantly affected by the interferogram unwrapping error and other errors, and the calculation result of R is small ($-0.05$). With the increase in the difference scale, the distribution of the elevation difference increases significantly. When the difference scale is 5 km (Figure 8F), the phase difference is concentrated between the fitted value $\pm2$ rad, and the elevation difference is concentrated between $\pm1.5$ km. The influence of the error on the correlation coefficient R tends to decrease, and the R result becomes larger ($-0.36$). In this case, it is not appropriate to use the correlation coefficient R to select the $K_1$ value, thus we choose the $K_1$ value estimated by the minimum spatial scale.

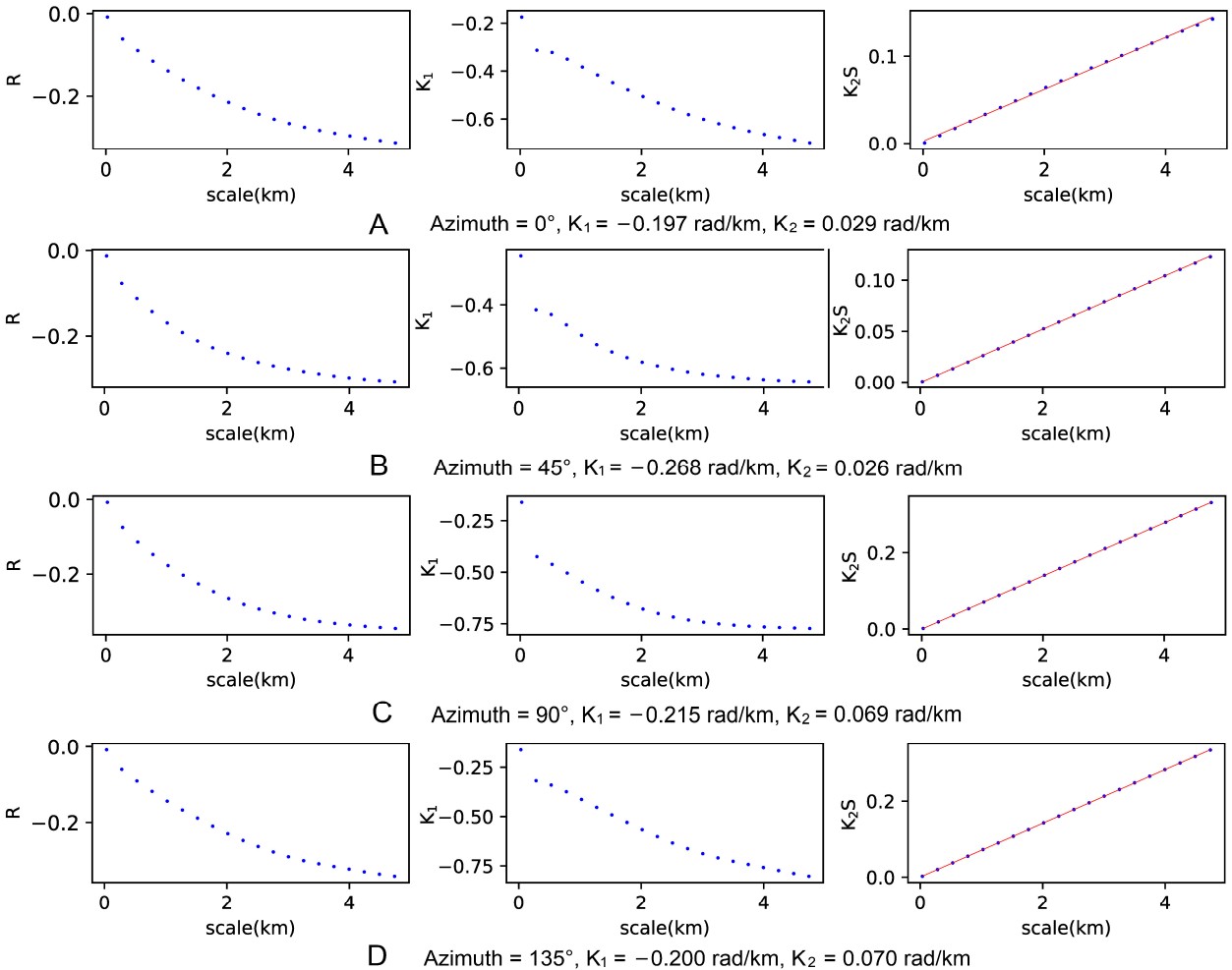

**Figure 7.** The values of the correlation coefficient R (column 1), $K_1$ (column 2), and $K_2S$ (column 3, blue dots, the red line is the fitting line of the blue dots) after the difference of the interferogram in four directions, and the final $K_1$, estimated according to the method described in the text. The maximum absolute value of $K_2$ is 0.070 rad/km (**D**), the corresponding azimuth is 135°, and the related $K_1$ is −0.200 rad/km. The $K_2$ values of other azimuth (**A–C**) are all less than 0.070 rad/km.

We used three methods to estimate the atmospheric phase delay. The first was the full interferogram–topography correlation approach, by which the estimated $K_1$ value was −0.22 rad/km. The second was the BP approach, by which the estimated $K_1$ value was −0.776 rad/km. The third was our MSSD approach, by which the estimated $K_1$ value was −0.20 rad/km. The $K_2$ value of the ramp estimated by our method was 0.07 rad/km, and the direction of the ramp was 135°. Next, we used the estimated transfer functions to correct the original interferogram. By using the $K_1$ parameters estimated from the full interferogram–topography correlation approach and BP approach, it still had an area of significant change from northwest to southeast (Figure 9B,C). By using the $K_1$ and $K_2$ parameters, estimated through the proposed approach, the phase distribution of the corrected interferogram acquired more uniform, and even a significant portion of the gradient was reduced (Figure 9C). The correlation coefficients between the corrected interferogram and topography were the criteria for the effectiveness of the chosen approach. Accordingly, the most suitable correction technique was the one that reduced this correlation significantly. To demonstrate the advantage of our approach, we divided the interferogram into nine sub-regions and then calculated the correlation coefficients of these regions, respectively. Figure 9B–D show the nine sub-regions, and the nine obtained correlation coefficients are illustrated in Figure 10 and Table 2. In theory, the better the correction effect is the smaller the absolute value of the correlation coefficients of each sub-region. It can be seen that the

absolute values of the correlation coefficient corrected by our approach in seven out of the nine sub-regions were smaller than those of the full interferogram–topography correlation approach and BP approach.

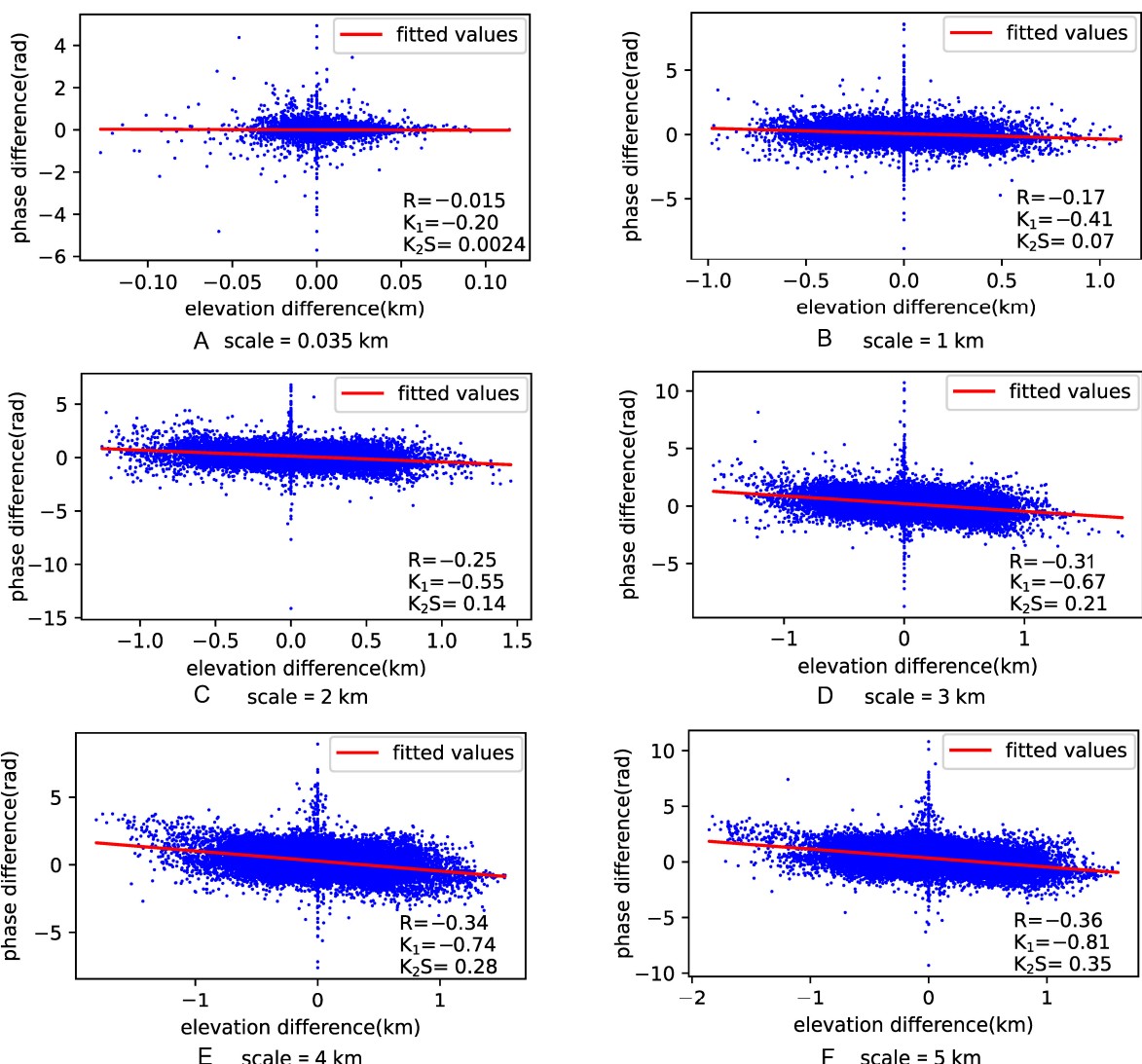

**Figure 8.** Scatter plots (phase difference vs. topographic height difference, blue dots) in various spatial scales in the Sierra Nevada Mountains with an azimuth of $135°$. This comparison shows the influence of error on correlation coefficient R in different scales.

**Table 2.** Correlation coefficients of 9 sub-regions.

| Sub Region | 0 | 1 | 2 | 3 | 4 | 5 | 6 | 7 | 8 |
|---|---|---|---|---|---|---|---|---|---|
| Original | −0.878 | −0.416 | 0.391 | −0.545 | −0.111 | 0.183 | 1.808 | 0.234 | 0.153 |
| Full-Igram | −0.743 | −0.280 | 0.532 | −0.406 | 0.024 | 0.319 | 2.030 | 0.374 | 0.284 |
| BP | −0.710 | −0.249 | 0.513 | −0.416 | 0.058 | 0.345 | 2.589 | 0.361 | 0.352 |
| MSSD | −0.128 | 0.044 | 0.528 | −0.052 | 0.026 | −0.001 | 2.151 | −0.004 | −0.248 |

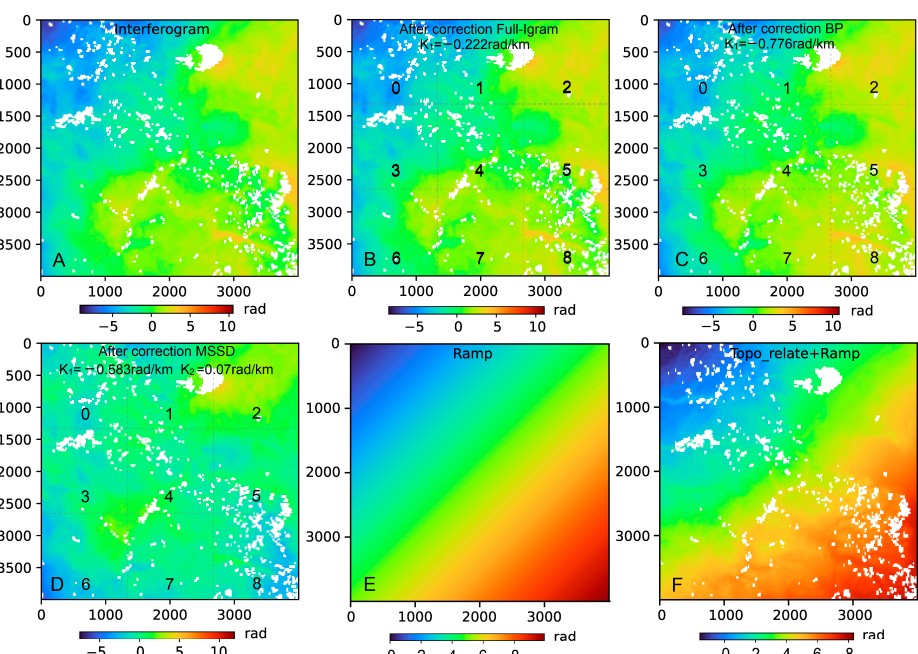

**Figure 9.** Comparison between the interferograms in the Sierra Nevada mountains before and after correction using the MSSD approach; (**A**) the original interferogram (**B**) the corrected interferogram obtained through the full interferogram–topography approach (**C**) the corrected interferogram obtained through the BP approach and (**D**) that obtained by MSSD; (**E**,**F**) the ramp component and the sum of topography-related component and ramp component, both acquired by the MSSD approach. Notice that the phase gradient in (**D**) is reduced after correction. Correlation coefficients of 9 sub-regions(0–8) are shown in Table 2.

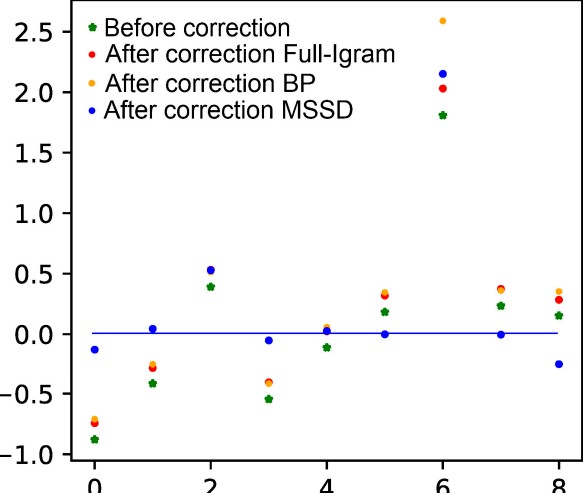

**Figure 10.** Correlation coefficients of 9 sub-regions; only the absolute values of the coefficients corrected by MSSD in the sub-regions 2, 4, and 6 are larger than those corrected by the full interferogram–topography correlation approach or BP approach (Table 2).

### 4.2. 2016 Menyuan Earthquake

An Mw = 5.9 earthquake struck Menyuan county, Qinghai (101.641°E, 37.67°N) on 21 January 2016. A moment–tensor solution from teleseismic data suggests that the Menyuan earthquake occurred on a 43° southern dipping thrust fault at about 10 km depth with a strike of 134° [35,36]. The hypocenter was located at the intersection of the Lenglongling fault and the Tuolaishan fault. This region is one of the most tectonically active areas [37] (Figure 11).

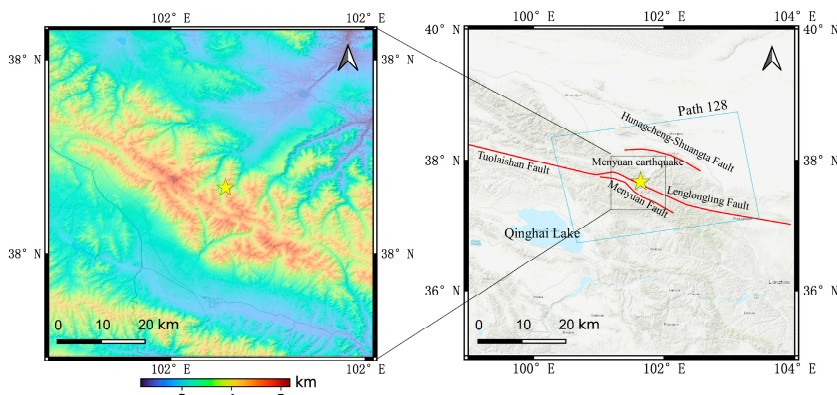

**Figure 11.** Reference map of the 21 January 2016 Menyuan Earthquake superimposed on topographic relief, with maximum elevation up to 5.17 km. The star shows the location of the 2016 Menyuan event. The red lines denote the active faults. The blue frames are the coverage of the Sentinel-1A data.

The coseismic deformation due to the 2016 Menyuan earthquake was mapped using the ascending track of the Sentinel-1A TOPS mode (path 128). Next, the ascending coseismic interferogram was generated from 13 January 2016 to 6 February 2016. The method and parameters of obtaining the interferogram were the same as those of the Sierra Nevada Mountains case. The temporal and spatial baselines were relatively small (15 m), and limited vegetation coverage existed in the epicenter region. Thus, the coherence was high, and the low-coherence points with coherence of less than 0.2 were masked. The interferogram is shown in Figure 12A, which clearly indicates a complex phase in this interferogram and a ramp from north to south. The patterns of the earthquake epicenter were smooth and distinct. Moreover, the interferogram made from these two SAR scenes was assumed to be dominated by coseismic deformation signals. In addition, we used the Okada elastic dislocation model [38] to construct the fault plane and then removed the modeled displacement field from the interferogram.

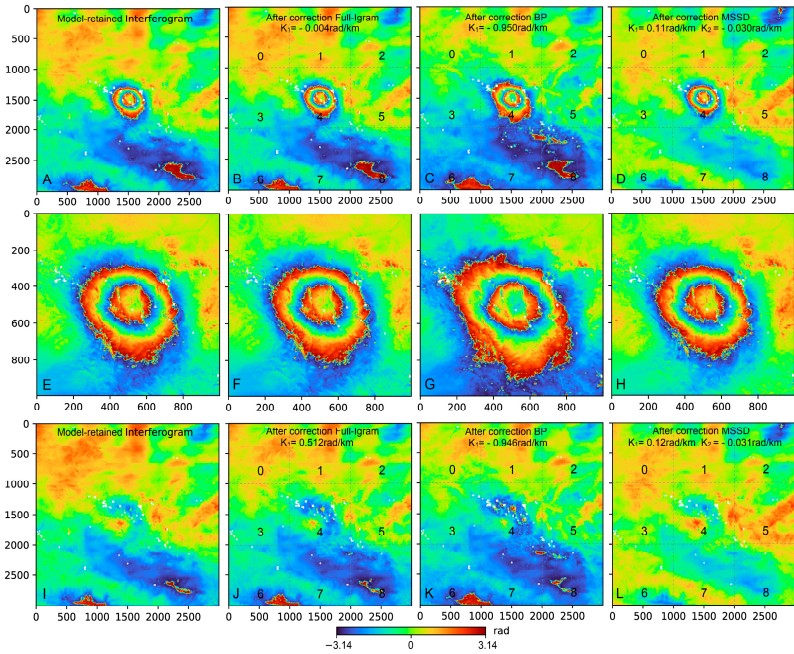

**Figure 12.** Comparison between the interferograms (wrapped) in the 2016 Menyuan coseismic displacement example before and after correction using the MSSD approach; (**A**–**D**) the model-retained interferograms before and after correction; (**E**–**H**) the local view of the epicenter area; (**I**–**L**) the model-removed interferograms before and after correction.



Our estimated $K_1$ values with the model-retained and model-removed interferograms using the proposed approach were very close: 0.11 and 0.12 rad/km (Figure 12). The $K_2$ values estimated by our method were $-0.030$ and $-0.031$ rad/km, respectively. In comparison, the $K_1$ values calculated by the full interferogram–topography correlation approach were $-0.004$ and 0.512 rad/km, and the $K_1$ values calculated by the BP approach were $-0.950$ and $-0.946$ rad/km. According to the findings, the derived $K_1$ and $K_2$ values by our approach were more stable. The phase gradient in the interferogram clearly decreased after correcting both the topographically correlated tropospheric and ramp signals with the $K_1$ and $K_2$ values (Figure 12D,L). Eventually, the full interferogram–topography correlation approach and BP approach reduced less gradients than the MSSD correction method. In the local view of the epicenter region (Figure 12E,F), the boundary of the earthquake region after atmospheric delay correction in our method was clearer than that in the other two methods.

We also calculated the correlation coefficients of nine sub-regions of the interferograms (Figure 13, Tables 3 and 4). In the model-retained interferogram, the correlation coefficients corrected by the MSSD method in eight sub-regions were better than the full interferogram–topography correlation approach. In contrast, the correlation coefficient in sub-region 4 could not be correctly estimated due to the influence of seismic deformation. In the model-removed interferogram, the correlation coefficients corrected by our approach in seven sub-regions were better than the full interferogram–topography correlation approach. In the two interferograms, the correlation coefficients of the seven sub-regions corrected by MSSD method were better than those of BP method.

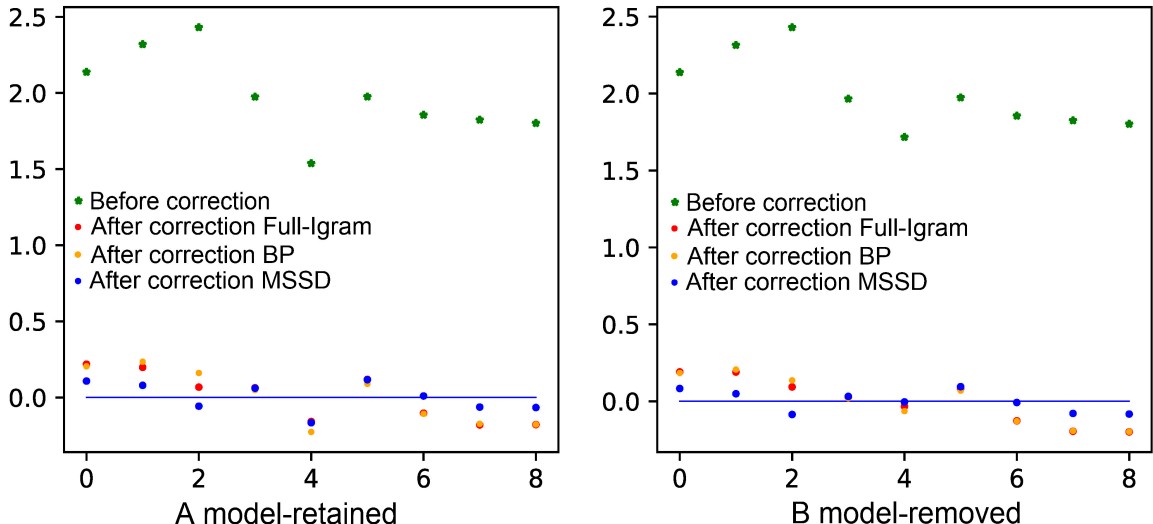

**Figure 13.** Correlation coefficients of 9 sub-regionss: (**A**) model-retained interferogram; (**B**) the model-removed interferogram.

**Table 3.** Correlation coefficients of 9 sub-regions before model-removed.

| Sub Area | 0 | 1 | 2 | 3 | 4 | 5 | 6 | 7 | 8 |
|---|---|---|---|---|---|---|---|---|---|
| Orignal | 2.138 | 2.320 | 2.431 | 1.975 | 1.538 | 1.976 | 1.856 | 1.824 | 1.802 |
| Full-Igram | 0.219 | 0.198 | 0.068 | 0.064 | $-0.158$ | 0.119 | $-0.103$ | $-0.180$ | $-0.178$ |
| BP | 0.205 | 0.235 | 0.161 | 0.049 | $-0.226$ | 0.090 | $-0.107$ | $-0.173$ | $-0.177$ |
| MSSD | 0.108 | 0.080 | $-0.057$ | 0.061 | $-0.166$ | 0.117 | 0.010 | $-0.063$ | $-0.066$ |

**Table 4.** Correlation coefficients of 9 sub-regions after model-removed.

| Sub Area | 0 | 1 | 2 | 3 | 4 | 5 | 6 | 7 | 8 |
|---|---|---|---|---|---|---|---|---|---|
| Orignal | 2.138 | 2.315 | 2.430 | 1.966 | 1.717 | 1.974 | 1.855 | 1.825 | 1.802 |
| Full-Igram | 0.191 | 0.190 | 0.093 | 0.027 | −0.033 | 0.082 | −0.127 | −0.195 | −0.199 |
| BP | 0.184 | 0.206 | 0.136 | 0.019 | −0.064 | 0.069 | −0.129 | −0.193 | −0.198 |
| MSSD | 0.083 | 0.049 | −0.086 | 0.032 | −0.004 | 0.095 | −0.008 | −0.079 | −0.083 |

## 5. Discussion and Conclusions

In this study, a simple approach using multi-scale spatial difference was used to analyze and correct the vertical stratification component delays. In this approach, the atmospheric phase delays were decomposed into the uncertainties of the vertical stratification component, ramp component, and other turbulent components. Next, we used our approach to estimate the transfer parameters of vertical stratification component delays and ramp components. The main idea of the approach is that the transfer function of the vertical stratification component delays will not change after the difference, while the difference value of ramp component will increase with the increase in spatial scale. In the actual algorithm implementation, we conducted multi-scale spatial differences in four directions, which may have had little influence on the $K_1$ and $K_2$ estimations, but we assume that can be ignored. The results of synthetic testing with various turbulence signals and ramps, along with a real interferogram encompassing the Sierra Nevada Mountains and the 2016 Menyuan earthquake, revealed that our method could more precisely estimate the transfer function. In the synthetic test, the multi-scale spatial difference approach offered a satisfactory insensitivity to approximate linear phase gradient, which had a good effect when there were orbital errors or long-wavelength scale signals. The actual example in the Sierra Nevada Mountains showed that it was more reasonable to choose the $K_1$ value estimated by the minimum spatial scale than to use the correlation coefficient due to the influence of noise and unwrapping error. The example of the Mengyuan earthquake showed that our method was more robust to deformation signals.

Our method assumed that the ramp signal was linear and independent of elevation. The usefulness of this strategy has to be carefully addressed if the ramp is not roughly linear. The approach proposed in our study was to obtain the optimal global transfer functions of the interferogram. In the analysis of real interferograms, due to the influence of local turbulence, after the multi-scale spatial difference approach, this may lead to the over-correction of short-wavelength signals. Therefore, we could not correct turbulence at all scales.

In conclusion, our multi-scale spatial difference approach should be considered a fast and handy tool when another method is not available. Still, it cannot cure all challenges posed by tropospheric delays.

**Author Contributions:** G.H. led the research work, Z.Y. designing the experiments and writing the first draft; Y.H. and Z.Z. contributed to experiment implementation; C.Z. and G.Z. contributed to paper writing and revision. All authors have read and agreed to the published version of the manuscript.

**Funding:** This research was funded by National Key R&D Program of China, grant number 2022YFB3902605, and The Fund of Beijing Key Laboratory of Urban Spatial Information Engineering, grant number 20220104.

**Data Availability Statement:** The Sentinel-1 data used in this study are downloaded from the European Space Agency (ESA) through the ASF Data Hub website https://search.asf.alaska.edu/ (accessed on 23 June 2022).

**Acknowledgments:** All Sentinel-1 data were obtained from the European Space Agency. The interferogram used in this study was generated using the ISCE software. We thank two anonymous reviewers for their detailed and thoughtful comments.

**Conflicts of Interest:** The authors declare no conflict of interest.

## Appendix A

In Figure A1, let the ramp gradient direction of the interferogram be the $OB$ direction, and then its corresponding $K_2$ value is:

$$K_2 = tan(\beta) = \frac{AB}{OA} \tag{A1}$$

while the approximate calculation direction is $OB'$, the estimated gradient value $K_2'$ is:

$$K_2' = tan(\beta') = \frac{A'B'}{OA'} \tag{A2}$$

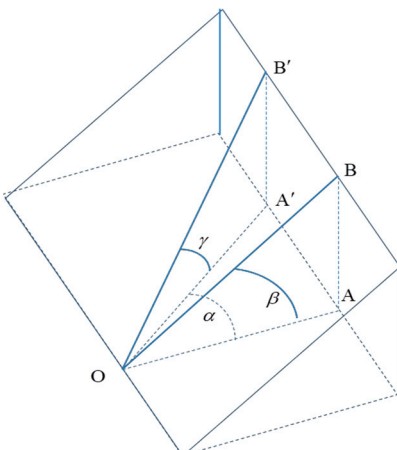

**Figure A1.** Error caused by approximate calculation of $K_2$ direction.

Let the angle between $OB$ and $OB'$ in the horizontal projection be $\alpha$, and then we can get:

$$\frac{K_2'}{K} = \frac{A'B'}{OA'} \frac{OA}{AB} \tag{A3}$$

Since $A'B' = AB$, Formula (3) can be sorted as follows:

$$\frac{K_2'}{K} = \frac{OA}{OA'} = cos(\alpha), \tag{A4}$$

or

$$K' = Kcos(\alpha). \tag{A5}$$

The maximum $\alpha$ is $\pi/8$, and the error of $K_2'$ is:

$$K_2' = K_2 cos(\pi/8) \approx 0.92K_2 \tag{A6}$$

Then, the estimated error of $K_2$ is $0.08K_2$, which can be ignored.

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
