# Peer review of "A Multi-Scale Spatial Difference Approach to Estimating Topography Correlated Atmospheric Delay in Radar Interferograms"

_remotesensing, doi:10.3390/rs15082115_

Round 1
Reviewer 1 Report
The paper "A multi-scale spatial difference approach to estimating topography correlated atmospheric delay in radar interferograms" proposed a new approach to estimate the transfer function of vertical stratification phase delays and the transfer function of phase ramp. The structure of this paper is clear, the experiments are sufficient. Generally, this paper could be accepted, whereas there are still some issues to be explained.
1.Page3.
In section part 2, it is said that “Turbulence signals are correlated in a short range (a few km),” Is there any research that proves or supports this claim,please give the basis?
2.Page11
In section part 3, it is said that “The size of the deviation is proportional to the amplitude of the ramp, in the same way, that the direction of deviation is related to the direction of the ramp.” Why the direction of deviation is related to the direction of the ramp?
3.Page12
In section part 4, in Figure 6D, it is said that “the overlay area in the interferogram.” To illustrate this statement, the overlay area can be graphically shown here.
4.Page14
In section part 4, it is said that “When the difference scale is 0.035km (Figure 8A), the phase difference is concentrated between the fitted value±1rad, and the elevation difference is concentrated in the range ±0.05km. The correlation coefficient R is significantly affected by the interferogram unwrapping error and other errors, and the calculation result of R is small (- 0.05). With the increase of the difference scale, the distribution of the elevation difference increases significantly. When the difference scale is 5km (Figure 8F), the phase difference is concentrated between the fitted value±2rad, and the elevation difference is concentrated between±1.5km.” How are these figures obtained?
5.Page15
In Figure 8A, there are two horizontal lines in the picture. What is the meaning of these two horizontal lines?
6.Page20
In section part 5, it is said that” In contrast, the ramp component will raise with the increase of scale, which is the fundamental principle of our approach.” Make sure that it is a correct expression.
Author Response
Dear Editor,
Thank you very much for the opportunity to submit a response of the manuscript (remotesensing-2275352).
We are grateful for all the comments from your reviews, which are very constructive and helpful. In the revised version, we have tried our best to improve the manuscript by incorporating all your comments into consideration including various enhancements in readability and corrections of Figures.
Following this letter are your comments with our responses including how and where the text was modified. Changes made in the manuscript are marked using track changes (with yellow highlighting indicating changes). Please see the attachment. The revision has been developed in consultation with all coauthors, and each author has given approval to the final form of this revision.
Kind Regards
Zhigang Yu

Reviewer 2 Report
The authors should concentrate on the purpose of the proposed method and show the advantages of the proposed method compared with the main correction methods in the test, such as being more precious and accurate in elevation estimation or deformation estimation.

Author Response

(The authors gave the same response as above.)

Round 2
Reviewer 2 Report
Thanks for the reply. I have read the revision. There are some mistakes in the font type and size. Please check them carefully. The other parts are better.
Author Response
Dear Editor,
Thank you very much for the opportunity to submit a response of the manuscript (remotesensing-2275352). We are grateful for all the comments from your reviews, which are very helpful. In the revised version, we have tried our best to improve the manuscript by your comments.We carefully reviewed the references and confirmed that all references were relevant to the article.
Following this letter are your comments with our responses including how and where the text was modified. Changes made in the manuscript are marked using track changes (with red highlighting indicating changes). Please see the attachment. The revision has been developed in consultation with all coauthors, and each author has given approval to the final form of this revision.
Kind Regards
Zhigang Yu
